# Engineering and Purification of Microcin C7 Variants Resistant to Trypsin and Analysis of Their Biological Activity

**DOI:** 10.3390/antibiotics12091346

**Published:** 2023-08-22

**Authors:** Guangxin Yang, Lijun Shang, Lu Liu, Zeqiang Li, Xiangfang Zeng, Xiuliang Ding, Jinxiu Huang, Shiyan Qiao, Haitao Yu

**Affiliations:** 1State Key Laboratory of Animal Nutrition and Feeding, Ministry of Agriculture Rural Affairs Feed Industry Centre, China Agricultural University, Beijing Bio-Feed Additives Key Laboratory, Beijing 100193, Chinashanglijun@cau.edu.cn (L.S.); liulu@cau.edu.cn (L.L.); qiaoshiyan@cau.edu.cn (S.Q.); 2Luzhou Modern Agriculture Development Promotion Center, Luzhou 646000, China; 3Shanghai Menon Animal Nutrition Technology Co., Ltd., Shanghai 201807, China; lizeqiang0521@126.com; 4Chongqing Academy of Animal Science, Rongchang, Chongqing 402460, China; dingxiuliang@swu.edu.cn (X.D.); huangjinxiu@swu.edu.cn (J.H.); 5National Center of Technology Innovation for Pigs, Rongchang, Chongqing 402460, China

**Keywords:** Microcin C7, trypsin-resistant variants, stability, antimicrobial activity, purification

## Abstract

Microcin C7 (McC) as a viable form of antimicrobial has gained substantial attention due to its distinctive antimicrobial activity, by targeting aspartyl tRNA synthetase. McC can be a potential solution against pathogenic microbial infections in the postantibiotic era. However, considering that degradation by digestive enzymes can disrupt the function of this peptide in the gastrointestinal tract, in this study, we attempt to design McC variants to overcome several barriers that may affect its stability and biological activity. The *mccA* gene encoding the McC peptide precursor was mutated and 12 new McC variants with trypsin resistance were found. The Yej^+^rimL^−^ strain was used as an indicator to determine the minimum inhibitory concentrations (MICs). The results showed that three variants, including R2A, R2T and R2Q, among 12 variants formed by the replacement of the second arginine of the McC peptide with different amino acids, were resistant to trypsin and had an outstanding antimicrobial ability, with MIC values of 12.5, 25, and 25 μg/mL, respectively. Taken together, our findings show that the engineering of the site-directed mutagenesis of McC significantly enhances McC trypsin resistance and maintains a great antimicrobial activity.

## 1. Introduction

Currently, most global antibiotics (73%) are used in the production of animals for human consumption [1]. The use of excessive antibiotics is currently the main strategy for dealing with pathogenic microbial infections in newborn and weaned animals, but the abuse of antibiotics can lead to antibiotic resistance among animal microbiota. From 2000 to 2018, the proportion of antimicrobial agents for which the resistance was greater than 50% increased from 0.15 to 0.41 in chickens, and the proportion increased from 0.13 to 0.34 in pigs [2]. In the postantibiotic era, the prohibition of the use of antibiotics will have serious impacts on the efficiency of animal husbandry production in China. Therefore, developing antibiotic substitutes and addressing the core scientific issues are urgent needs with regard to the sustainable development of China’s animal husbandry. On the other hand, for the novel peptides we engineered, the important first step is that the peptides retain their stability and biological activity in such drastic conditions present in the gastrointestinal context.

The members of a subclass of antimicrobial peptides (AMPs), called bacteriocins, are effective only against a narrow spectrum of bacteria closely related to the producer. Increasing attention has been paid to bacteriocins as antimicrobial compounds due to their narrow inhibitory spectrum [3]. Scientists have developed new antibiotics by identifying genes in pathogenic bacteria that are absent, or that lack homologs in humans. Aminoacyl-tRNA synthases (aaRSs) are considered promising targets [4]. Microcin C7 (McC) is a natural antimicrobial peptide produced by *Escherichia coli* (*E. coli*) cells carrying the mcc cluster and exerts antimicrobial activity against a wide range of Gram-negative bacteria, including *Escherichia*, *Klebsiella*, *Salmonella*, *Shigella*, and *Proteus* species [5,6,7]. Mature McC is covalently linked to the aspartic acid carboxyl group of the MRTGNAD heptapeptide through a phosphoramidate bond; the phosphate group is modified by a propylamine group and the N-terminal methionine is formylated (Appendix A).

After the formation of mature McC, McC enters the target bacteria. f-Met on the N-terminus of McC is recognized by the porin OmpF on the outer membrane of a target bacterial cell, and then by the ABC transporter YejABEF on the inner membrane of the cell. Novikova et al. concluded that yej mutations interfere with McC uptake and YejABEF is the only inner membrane transporter responsible for McC uptake in *E. coli* [8]. Once McC enters the target bacterial cell, processing begins. First, the N-terminal formyl group is removed by deformylase. Next, the resulting deformylated group is digested by one or more of three aminopeptidases—PepA, PepB, and PepN—and when the peptide bond between the sixth and seventh peptide residues is hydrolyzed, a nonhydrolyzable analog of aspartyl adenylate with a propylamino group is released [9]. The Trojan horse mechanism has attracted the interest of researchers, and large numbers of McC analogs have been produced. McC has, thus, become an excellent model for research on elegant and complex strategies to produce effective and selective drugs [10].

The heptapeptide precursor of McC, f-MRTGNAN, is encoded by the first gene in the mcc gene clusters, *mccA* [11]. Considering its protein nature, it is suitable for gene-based peptide engineering [12]. Kazakov et al. first applied site-directed mutagenesis on the *mccA* gene, systematically replacing the second to seventh codons of *mccA* encoding 19 standard amino acids for a total of 114 mutations [13]. Twenty-eight variants did not significantly change the biological activity of the peptide, and four substitutions (R2Y, R2M, A6M, and A6F) induced a loss of antimicrobial activity. These findings indicate that arginine at position 2 may be a recognition molecule taken up by the cell and that alanine at position 6 is responsible for processing. However, this method could not be used to study the role of the first methionine of the MccA peptide since this residue is necessary for translation initiation. The unique mechanism of McC makes it an attractive model for the design of other aminoacyl-tRNA synthetase analogs, but the saturation of the last position of the seven peptides at the molecular level has not been successfully synthesized. Vijver et al. changed the seventh amino acid through total chemical synthesis and obtained three McC-like compounds in which a terminal aspartic acid, glutamic acid, or leucine was connected to adenosine through a nonhydrolyzed sulfamic acid bond [14]. Although these analogs lacked the N-terminal formyl group, they were still active and retained the mechanism of wild-type McC, targeting AspRS, GluRS, and LeuRS. The effect of the length on McC transport has been studied via the chemical synthesis of McC analogs of different lengths, and the results have shown that McC-shortening amino acids significantly reduce biological activity by affecting yejABEF-mediated transport [15]. Notably, Bantysh et al. used a recombinant MccB protein to prepare McC analogs with different peptide chain lengths and revealed that extension of the N-terminus of the MccA heptapeptide does not affect the adenosylation of the MccB protein, and some of the N-terminally extended McC analogs exhibited increased antimicrobial activity [16]. 

The wild-type McC polypeptide sequence is MRTGNAN, which can undergo cleavage by trypsin at the R2 position. This feature makes McC unsuitable for intestinal administration and is a hindrance to its use as a replacement for antibiotics. Therefore, in this study, we used a circular plasmid containing the wild-type mcc gene sequences as a template to construct trypsin-resistant McC variants through genetic engineering technology. An McC-variant HPLC purification method was established, and the antimicrobial effects of the variants were detected via the determination of the minimum inhibitory concentrations (MICs). We successfully screened variants from the variant library with low MIC values and resistance to trypsin. This represents the first example of the use of peptide engineering to create trypsin-resistant microcins.

## 2. Results

### 2.1. Production of the Mutated Peptides

Twenty-one variants based on the circular plasmid pWP40 containing the wild-type mcc gene sequences were constructed (Table 1). QuikChange site-directed mutagenesis was used to construct 17 variants, and Gibson assembly was used to construct four variants: R2T, R2F, RPT, and T3P (Appendix A). The sequences of all constructed variants were verified by DNA sequencing, and the amino acid sequences of all mutants are shown in Table 1. The supernatants obtained from the variant fermentation broth contained the variant proteins.

### 2.2. Production of the Mutated Peptide Activity and Protease Resistance of the McC Variants

To analyze the antimicrobial ability of the variants, we measured the inhibition zones of all 21 variants by the agar diffusion method [10]. The peptide moiety of McC was recognized by a YejABEf-peptide transporter. The rimL contains an acetyltransferase domain homologous to MccE which can detoxify and provide resistance to McC [17]. The overexpression of the *YejABEF* gene cluster and rimL gene knockout made the strain hypersensitive to McC. Therefore, the Yej^+^rimL^−^ strain was selected as the sensitive indicator bacterial strain.

Seven variants, R2W, R2C, R2P, R2N, R2E, T3P, and RPT, had no bacteriostatic effects on two indicator strains (Yej^+^rimL^−^ and K88). Five variants, R2Y, R2L, R2V, R2G, and R2D, were effective only on the Yej^+^rimL^−^ strain, and the diameters of the inhibition zones were 18.87, 14.40, 21.34, 18.86, and 17.71 mm, respectively. Except for R2F and R2Q, the remaining seven variants exhibited greater bacteriostatic activity against the Yej^+^rimL^−^ indicator strain than against the K88 indicator strain (Table 2).

To test whether the variants were tolerant to trypsin or other digestive enzymes, the amount of the remaining McC and its analogues remaining was determined by RP-HPLC after trypsin, pepsin, and chymotrypsin treated for 3 h (Appendix A), and the inhibition zones of trypsin-treated variants were tested using the same method (Table 2). When the McC-sensitive strain Yej^+^rimL^−^ was used as an indicator, the inhibitory zones diameters of all variants were slightly smaller after trypsin treatment than before trypsin treatment. Fourteen variants with antimicrobial effects on the Yej^+^rimL^−^ strain were treated with trypsin, and twelve variants exhibited antibacterial effects, as follows (listed in order of descending inhibitory zone diameter): R2Q, R2A, R2T, R2S, R2H, R2I, R2Y, R2V, R2M, R2G, R2D, and R2L. The R2L and R2K variants no longer had bacteriostatic activity after trypsin treatment, and wild-type McC also lost its antimicrobial activity after trypsin treatment. Although the other mutants were no longer antimicrobial-active, they also showed resistance to trypsin. These results are consistent with the digestive enzyme experiment (Appendix A). In addition, among these mutants, T3P and RPT mutants were no longer degraded by trypsin, as previously reported by linking proline after arginine to avoid cleavage by trypsin [18]. Although the other mutants were no longer antimicrobial-active, they also showed resistance to trypsin. When K88 was used as indicator strain, five trypsin-treated variants retained their antimicrobial activity, namely, R2Q, R2A, R2H, R2T, and R2S (in descending order).

### 2.3. Optimization of HPLC Purification Conditions

To find the separation peaks of the variants, wildtype McC and blank control fermentation broth samples were tested by HPLC. The separation peaks with obvious differences (those with retention times of 11 min and 17 min) were purified and the antimicrobial activity was determined. The results of the agar diffusion test showed that the supernatant obtained at two different time points formed circular antimicrobial areas. Therefore, the molecular weight was further analyzed by mass spectrometry, and the separation peak at approximately 17 min was confirmed to be that of the desired purified McC. Since the mutations in this study involved only one amino acid and had few effects on the properties of the peptides, we predicted that the separation peaks of the variants should fluctuate around approximately 17 min according to their hydrophobicity. The results showed that the retention times of the variants varied from 16 to 19 min according to hydrophobicity. The purity of the purified variants was determined by HPLC to be greater than 80% (Table 3).

To maximize the collection of single-peak samples and to perform activity tests, the 12 variants that retained antimicrobial activity against the Yej^+^rimL^−^ strain after trypsin treatment were purified by HPLC, and the B flow ratio was adjusted from 10% to 17%. After the target peak reached a better resolution, the cycle was collected (Appendix A).

### 2.4. Characterization of the McC Variants

To confirm that the purified substances were the desired variants, we verified their molecular masses by mass spectrometry. The spectra were in line with the theoretical molecular masses (Table 4).

The mass spectra of five variants that retained antimicrobial activity against K88 after trypsin treatment were as follows (Figure 1): R2A, [M + H]^+^ = 1092.30 and [M + 2H]^2+^ = 546.70; R2S, [M + H]^+^ = 1108.30 and [M + 2H]^2+^ = 554.80; R2H, [M + H]^+^ = 1158.40 and [M + 2H]^2+^ = 579.70; R2T, [M + H]^+^ = 1122.30 and [M + 2H]^2+^ = 561.80; and R2Q, [M + H]^+^ = 1149.40 and [M + 2H]^2+^ = 575.30.

To accurately characterize the bacteriostatic abilities of the variants, the MICs of the pure McC variants toward the McCsensitive strain Yej^+^rimL^−^ were determined. The MICs of all variants were lower than that of wild-type McC. From smallest to largest MIC, the above five variants were in the following order: R2A, R2T, R2Q, R2S, and R2H. The MIC values of R2T and R2Q were both 25 μg/mL (Table 5). Although the diameter of the inhibitory zone of R2M was smaller than those of the above five variants, the bacteriostatic ability of R2M was second only to R2T/R2Q, and the MIC was 50 μg/mL.

## 3. Discussion

The McC as a new generation of antimicrobials is now receiving increased attention because of its antimicrobial activity against a wide range of Gram-negative bacteria, including *Escherichia*, *Klebsiella*, *Salmonella*, *Shigella*, and *Proteus* species [5,6,7]. However, inactivated or degraded proteolytic enzymes in the stomach and the small intestine, such as pepsin, trypsin, and chymotrypsin, need to be considered before the application of McC or other microcins in different fields [19,20,21]. Yu et al. has demonstrated that microcin J25 was stable in the gastrointestinal tract in vitro and in vivo but completely degraded upon exposure to chymotrypsin [22]. Therefore, due to insufficient data on the trypsin-resistant McC, here, we focus on the most recent trends relating to the trypsin resistance of microins, and their engineered and biological activity.

In this study, we constructed trypsin-resistant McC variants and found that the mutation of McC affects McC antimicrobial activity to different degrees. To construct the McC trypsin-resistant variants, we adopted three strategies: (1) the mutation of the Arg at position 2 on the McC peptide to 19 other amino acids; (2) the addition of the protective amino acid Pro after the Arg at the second position; and (3) the mutation of Thr at the third position of the McC peptide to Pro. Twenty-one variants were constructed in total and treated with trypsin. The agar diffusion test results showed that 12 variants were resistant to trypsin and had antimicrobial activity against Yej^+^rimL^−^. To accurately determine the antimicrobial ability of the variants, we established an HPLC purification method for McC, and assessed the MICs of the pure McC variants against the McC-sensitive strain Yej^+^rimL^−^. Our findings showed that replacement of the second position of the McC peptide with different amino acids had different degrees of impact on antimicrobial activity. Among the constructed 21 variants, three variants, R2A, R2T, and R2Q, had a nearly ideal bacteriostatic effect on the indicator bacterial strain Yej^+^rimL^−^. The variants had the advantage of being resistant to trypsin and exhibited a strong antimicrobial activity. 

The results of the agar diffusion test showed that the diameters of the inhibitory zones of the variants were smaller than that of wild-type McC, indicating that the second Arg of the McC peptide has an important effect on the antimicrobial activity of McC. Similar to previous studies, our study revealed that the exclusion of Arg from the peptide significantly reduced the activity of McC [15]. However, in contrast to our study, the study in which the second to the seventh codons of *mccA* were replaced with 19 standard amino acid codons revealed that only five variants with Arg replacement were bacteriostatic: R2A, R2S, R2H, R2W, and R2Y. These differing results likely resulted from the different expression in the cells or from the use of different indicator bacteria [13]. 

McC peptides play important roles in the exertion of antimicrobial activity. They are closely related to the uptake of McC variants through the YejABEF inner membrane transporter. The result shows that several amino acids, that are mainly non cationic ones (A, S, H, I, M, T, Q, Y, L, V, G, and D), can replace Arg2 without suppressing the activity. Although the MICs of the variants were higher than that of the wild-type McC (1.56 μg/mL), three variants, R2A, R2T, and R2Q, showed outstanding antibacterial ability among the 12 variants, with MIC values of 12.5, 25, and 25 μg/mL, respectively. This finding confirms that the different amino acid substitutions had different effects on antibacterial activity and McC recognition by YejABEF transporters. All mutations showed reduced effectiveness. The sequence of the inhibitory ability of variants obtained by the agar diffusion test and MIC test is different. This phenomenon suggested that the levels of the variants in the fermentation broth were different and that purification of the variants was necessary to determine the MIC. Additionally, in our study, for the RPT analogue, a peptide chain length of eight peptides, we inserted a new amino acid after the second AA of McC. The increase in the AA chain length causing McC to lose its antimicrobial activity was observed compared with other shorter peptide chains (7 AA). Notably, Vondenhoff et al. have found that, compared with wild-type McC, McC analogs containing one to six amino acids exhibited higher MIC values against *E. coli* K-12 (BW28357), which is similar to the trends of the MIC results in our study. These findings suggested that the peptide chain length may have effects on 238 YejABEF transporter recognition and antimicrobial activity [10]. In addition, variants based on the lantibiotic Pep5 have been studied, and the results have shown that the replacement of Ala at position 19 with Cys decreases the antimicrobial activity of the peptide, although it enhances the stability of chymotrypsin hydrolysis [23].

When the Yej^+^rimL^−^ strain was used as an indicator bacterial strain, the diameter of the inhibition zones of McC was 5.68 mm larger than that of chloramphenicol due to the greater effectiveness of the bacteriostatic components of McC. Chloramphenicol is an effective antibiotic; hence, it had the inhibition zones with the largest diameter when the K88 strain was used as the indicator bacterial strain. As mentioned above, YejABEF has a preference for peptides, and MR residues are particularly important [15]. Our results indicated that five variants, R2W, R2C, R2P, R2N, and R2E, lacked antimicrobial activity, probably because YejABEF could not recognize the peptides [13]. Another reason for this finding may be that McC also has a self-immune system. Because the induction capacity was unchanged, the variant was not depleted by the self-protection system [24,25]. The lack of activity may have also been due to a functional defect, insufficient production, or both. The five variants R2Y, R2L, R2V, R2G, and R2D had antimicrobial activity only against the Yej^+^rimL^−^ strain, which may indicate that production was insufficient.

Placing Pro at the C-terminus of Arg and Lys can block the trypsin-mediated cleavage at Arg and Lys [18]. In this study, the T3P and RPT variants exhibited resistance to trypsin, and the results showed that these two variants lacked antimicrobial activity. Likewise, Kazakov et al. demonstrated that the third amino acid of McC is highly conserved, and only mutations to Cys, Leu, and Thr were successful. Interestingly, the results of that study also revealed that mutations of amino acid positions 2 to 6 of wild-type McC to Pro were unsuccessful [13]. After trypsin treatment, the variant R2F no longer had bacteriostatic activity, which may have been due to an insufficient quantity or to poor bacteriostatic ability. R2K and wild-type McC also lacked bacteriostatic activity because Arg and Lys are trypsin-specific recognition sites, so they are cleaved by trypsin in simulated artificial intestinal fluid. Such cleavage induces the loss of antimicrobial activity.

## 4. Materials and Methods

### 4.1. Bacterial Strains, Plasmids, and Culture Conditions

The bacteria and plasmids used in this study are shown in Table 6. For construction of McC variants, the strains containing the corresponding McC variant plasmids were grown on 37 °C in LB medium supplemented with 100 μg/mL ampicillin. Host bacteria carrying McC and variant plasmids were grown in fermentation broth supplemented with 100 μg/mL ampicillin (2% yeast powder and 0.6% potassium dihydrogen phosphate). The Yej^+^rimL^−^ strain was grown in LB broth supplemented with 100 μg/mL ampicillin (Sloarbio) and 50 μg/mL kanamycin (Sloarbio).

### 4.2. DNA Preparation and Transformation

TianGen Plasmid Mini Rapid Extraction Kit was used to extract plasmid DNA. Plasmid DNA containing the McC variant was extracted using a TIANprep mini plasmid kit (Tiangen Biotech, Beijing, China). Variant plasmid DNA was transferred into *E. coli* top 10 competent cells according to the manufacturer’s instructions, and sequencing was performed by BGI (Huada Genomics Institute Co. Ltd., Shenzhen, China). *E. coli* MC4100 competent cells were prepared by the CaCl_2_ method. The variants verified by sequencing were transferred into MC4100 competent cells for expression.

### 4.3. Site-Directed Mutagenesis of the mccA Gene Encoding the McC Precursor Peptide

We use a circular pWP40 plasmid containing the wild-type mcc gene sequences as a template (Figure 2).

Primers containing mutation sites were designed according to the codon preference of *E. coli* (Table 7). Mutations were introduced into the *mccA* gene of the wild-type McC plasmid using QuikChange site-directed mutation method. Four variants, R2T, R2F, RPT, and T3P, were constructed using the Gibson assembly method. For example, to construct R2T, L-23F and L-23R were used to synthesize 1.4 kb fragments, and L-24F and L-24R were used to synthesize 7.4 kb fragments. Then, the two fragments were assembled with a Gibson mix. The other three variants were prepared in the same way. The enzymes Prime STAR and *Dpn* I were purchased from Thermo Scientific (Thermo Fisher Scientific, Waltham, MA, USA) and TaKaRa (Takara Biomedical Technology (Beijing) Co., Ltd., Beijing, China).

### 4.4. Fermentation

To express McC and its variants, host bacteria carrying the corresponding plasmids were cultured in 5 mL of fermentation broth supplemented with 100 μg/mL ampicillin at 37 °C, and the pH value was adjusted to 6.2 with HCl; after 12 h, the precultured cells were inoculated (2% *v*/*v*) into 40 mL of fermentation broth (pH = 6.2) and incubated at 220 rpm and 37 °C for 22 h.

### 4.5. Trypsin Treatment of the McC Variants

The fermentation broth was collected via centrifugation at 12,000× *g* at 4 °C for 10 min; the supernatants contained the McC variants. The supernatants and trypsin were mixed at a ratio of 9:1 (*v*/*v*; the final concentration of trypsin was 50 μg/mL) and incubated in a bath at 37 °C for 3 h. The pH was adjusted to 6.8. The pH value of the fermentation broth was measured before and after treatment, and an approximate pH value of 7 was considered normal.

### 4.6. Analysis of the McC Variants

The lyophilized supernatants were dissolved in deionized water, filtered through 0.22 μm aqueous-phase filter membranes, and purified by high-performance liquid chromatography (HPLC, Agilent Technologies Inc., Palo Alto, CA, USA) with C18 column (5 μm 9.4 × 250 mm, Agilent Technologies Inc., Palo Alto, CA, USA). The mobile phase was composed of 0.1% (*v*/*v*) trichloroacetic acid (solvent A) and 0.1% (*v*/*v*) trichloroacetic acid with 90% (*v*/*v*) acetonitrile (solvent B). A linear gradient from 95% A/5% B to 50% A/50% B was used with a flow rate of 1 mL/min, a sample volume of 10 μL, a column temperature of 35 °C, and a detection wavelength of 280 nm. Electrospray ionization mass spectrometry (ESI-MS) was used to compare the McC variant masses with the expected masses. The relationships between the relative abundances and the mass-to-charge ratios (m/z) of the ions were determined to confirm the identities of the experimentally produced McC variants.

### 4.7. Determination of Antimicrobial Activity and Minimal Inhibitory Concentrations

The fermentation supernatants and pure products were used to characterize the bacteriostatic effects of McC variants by the agar diffusion and microdilution methods. Yej^+^rimL^−^ and *E. coli* K88 indicator bacteria were applied. The serotype of *E. coli* K88 was O149:K91, K88ac, a clinically isolated strain, and provided by the College of Veterinary Medicine, China Agricultural University. The indicated bacteria were diluted to 1 × 10^5^ CFU/mL and added to plate medium cooled to 50 °C. Two hundred microliters of supernatants were added to each dish, and diffusion was performed at 4 °C for 1 h. After incubation at 37 °C for 24 h, the diameters of the inhibition zones were determined. One hundred microliters of 0.25 μg/mL chloramphenicol was used as a control. The MICs of the pure McC variants were determined by microdilution in a sterilized 96-well microplate. The McC variants were serially diluted 2 times in 96-well microplates with LB medium. The concentrations of the McC variants ranged from 0.125 to 100 µg/mL. Each well was inoculated with 10 μL of overnight bacterial culture at a concentration of 1.0 × 10^5^ CFU/mL and incubated at 37 °C for 24 h. A microplate reader was used to automatically observe changes in absorbance at 600 nm in order to monitor bacterial growth. A positive control group (including inoculated medium) and a negative control group (with medium only) were included. The MIC was determined as the lowest concentration of McC variant that inhibited bacterial growth (preventing an increase in the absorbance reading). All tests were performed in triplicate.

## 5. Conclusions

To summary, microcins have great potential for use as antimicrobial substitutes for antibiotics, but the sensitivity of microcins to digestive enzymes is still a problem that urgently needs to be solved. In this study, we have shown that the trypsin resistance of McC can be enhanced by site-directed mutagenesis and still keep a great antimicrobial ability. While we acknowledge that the trypsin-resistant variants described in this study exhibit lower antimicrobial activity than the original McC, the creation of these variants represents an important step forward and could find potential use in animal, food, and clinical applications. 

## Figures and Tables

**Figure 1 antibiotics-12-01346-f001:**
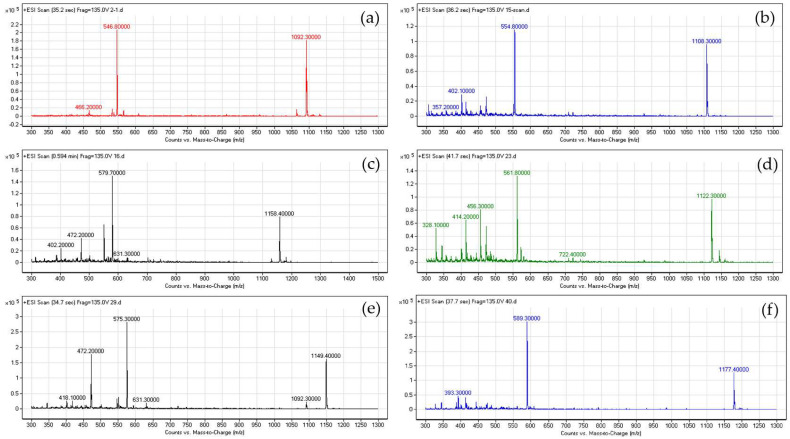
Mass spectra of McC variants: (**a**) mass spectrum of R2A, (**b**) mass spectrum of R2S, (**c**) mass spectrum of R2H, (**d**) mass spectrum of R2T, (**e**) mass spectrum of R2Q, and (**f**) mass spectrum of McC. Different colors in the mass spectra were autogenerated by the mass spectrometry software, which don’t imply any other details.

**Figure 2 antibiotics-12-01346-f002:**
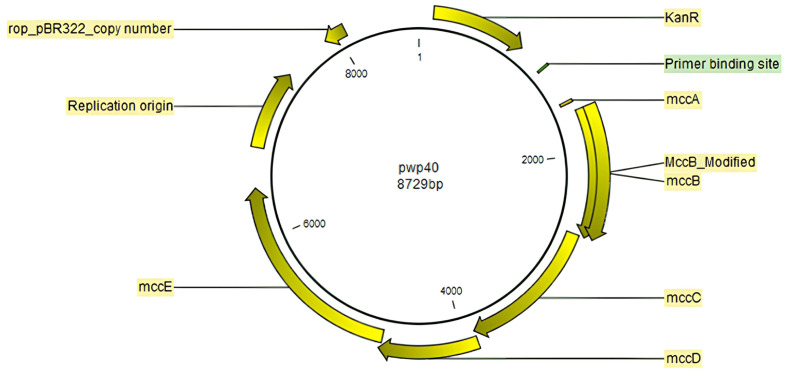
pWP40 plasmid information. pWP40 is a plasmid containing the wild-type mcc genes that are composed of pBR322 replicons, *mccA*, *mccB*, *mccC*, *mccD,* and *mccE* genes. Yellow: the composition of plasmid structure; Green: binding sites of mutant primers.

**Table 1 antibiotics-12-01346-t001:** Mutagenic oligonucleotides and primers used in mutagenesis.

Variant Plasmids	Mutagenic Oligonucleotides	Amino Acid Sequence	Primers
pLL14	R2A	MATGNAN	Z14f/Z14r
pLL15	R2S	MSTGNAN	Z15f/Z15r
pLL16	R2H	MHTGNAN	Z16f/Z16r
pLL17	R2W	MWTGNAN	Z17f/Z17r
pLL18	R2Y	MYTGNAN	Z18f/Z18r
pLL19	R2L	MLTGNAN	Z19f/Z19r
pLL20	R2I	MITGNAN	Z20f/Z20r
pLL21	R2V	MVTGNAN	Z21f/Z21r
pLL22	R2M	MMTGNAN	Z22f/Z22r
pLL23	R2T	MTTGNAN	L-23F/L-23R, L-24F/L-24R
pLL24	R2G	MGTGNAN	Z24f/Z24r
pLL25	R2C	MCTGNAN	Z25f/Z25r
pLL26	R2P	MPTGNAN	Z26f/Z26r
pLL27	R2F	MFTGNAN	L-23F/L-27R, L-27F/L-24R
WpLL28	R2N	MNTGNAN	Z28f/Z28r
pLL29	R2Q	MQTGNAN	Z29f/Z29r
pLL30	R2K	MKTGNAN	Z30f/Z30r
pLL31	R2D	MDTGNAN	Z31f/Z31r
pLL32	R2E	METGNAN	Z32f/Z32r
pLL34	RPT	MRPTGNAN	L-23R/L-34F, L-34R/L-24R
pLL35	T3P	MRPGNAN	L-23F/L-35F, L-35R/L-24R

**Table 2 antibiotics-12-01346-t002:** Diameters of the inhibition zones of McC variants.

Variants	Trypsin-Untreated Group		Trypsin-Treated Group	
	Yej^+^rimL^−^	K88	Yej^+^rimL^−^	K88
R2A	21.82	16.53	19.07	11.96
R2S	17.35	13.46	16.34	9.44
R2H	18.39	15.55	15.92	11.99
R2W	-	-	-	-
R2Y	18.87	-	14.93	-
R2L	14.41	-	0	-
R2I	18.36	10.18	15.67	-
R2V	21.34	-	14.62	-
R2M	21.57	12.37	14.31	-
R2T	21.74	15.27	17.23	11.68
R2G	18.86	-	14.15	-
R2C	-	-	-	-
R2P	-	-	-	-
R2F	13.05	14.34	10.95	-
R2N	-	-	-	-
R2Q	19.45	21.67	19.38	14.56
R2K	20.91	16.95	0	-
R2D	17.71	-	14.11	-
R2E	-	-	-	-
RPT	-	-	-	-
T3P	-	-	-	-
McC	29.12	20.25	-	-
Chloramphenicol	23.44	23.63	20.96	22.79

**Table 3 antibiotics-12-01346-t003:** HPLC purification information of variants.

Variants	Retention Time/min	Purity (%)
R2A	17.167	93.05
R2S	17.004	94.86
R2H	16.872	95.47
R2Y	18.591	92.30
R2L	19.425	81.38
R2I	18.166	92.51
R2V	18.151	81.74
R2M	19.622	88.21
R2T	17.418	93.19
R2G	17.253	98.06
R2Q	16.910	91.37
R2D	17.625	94.95
McC	17.198	84.38

**Table 4 antibiotics-12-01346-t004:** Molecular mass of McC variants.

Variants	Molecular Mass
R2A	1091.339
R2S	1107.339
R2H	1157.409
R2Y	1183.439
R2L	1133.419
R2I	1133.419
R2V	1119.399
R2M	1151.459
R2T	1121.369
R2G	1077.319
R2Q	1148.399
R2D	1135.349
McC	1176.449

**Table 5 antibiotics-12-01346-t005:** MICs of McC variants.

Variants	R2A	R2S	R2H	R2Y	R2L	R2I	R2V	R2M	R2T	R2G	R2Q	R2D	McC
MIC (μg/mL)	12.5	100	>100	>100	>100	100	50	50	25	>100	25	>100	1.56

Note: The Yej^+^rimL^−^ strain was used as indicator.

**Table 6 antibiotics-12-01346-t006:** Strains and plasmids.

Strains and Plasmids	Relevant Properity	Source
*E. coli* Top 10	Cloning strain	TianGen
*E. coli* MC4100	Expression strain	Our laboratory
Yej^+^rimL^−^	Knr and Ampr	Our laboratory
K88	/	Our laboratory
pWP40	Carrying the mcc genes, Knr	Our laboratory
pLL14	Knr	This paper
pLL15	Knr	This paper
pLL16	Knr	This paper
pLL17	Knr	This paper
pLL18	Knr	This paper
pLL19	Knr	This paper
pLL20	Knr	This paper
pLL21	Knr	This paper
pLL22	Knr	This paper
pLL23	Knr	This paper
pLL24	Knr	This paper
pLL25	Knr	This paper
pLL26	Knr	This paper
pLL27	Knr	This paper
pLL28	Knr	This paper
pLL29	Knr	This paper
pLL30	Knr	This paper
pLL31	Knr	This paper
pLL32	Knr	This paper
pLL34	Knr	This paper
pLL35	Knr	This paper

**Table 7 antibiotics-12-01346-t007:** Primers used for site-directed mutagenesis.

Primers ^1^	DNA Sequence (5′-3′)
z14f	GGAGGCGTAAAATGgctACTGGTAATGCAAAC
z14r	GTTTGCATTACCAGTagcCATTTTACGCCTCC
z15f	GGAGGCGTAAAATGagtACTGGTAATGCAAAC
z15r	GTTTGCATTACCAGTactCATTTTACGCCTCC
z16F	GGAGGCGTAAAATGcatACTGGTAATGCAAAC
z17f	GGCGTAAAATGtggACTGGTAATGC
z17r	GCATTACCAGTccaCATTTTACGCC
z18f	GGCGTAAAATGtatACTGGTAATGC
z18r	GCATTACCAGTataCATTTTACGCC
z19f	GGCGTAAAATGcttACTGGTAATGC
z19r	GCATTACCAGTaagCATTTTACGCC
z20f	GGCGTAAAATGattACTGGTAATGC
z20r	GCATTACCAGTaatCATTTTACGCC
z21f	GGCGTAAAATGgttACTGGTAATGC
z21r	GCATTACCAGTaacCATTTTACGCC
z22f	GGCGTAAAATGatgACTGGTAATGC
z22r	GCATTACCAGTcatCATTTTACGCC
z23f	GGCGTAAAATGaccACTGGTAATGC
z23r	GCATTACCAGTggtCATTTTACGCC
z24f	GGCGTAAAATGggtACTGGTAATGC
z24r	GCATTACCAGTaccCATTTTACGCC
z25f	GGCGTAAAATGtgtACTGGTAATGC
z25r	GCATTACCAGTacaCATTTTACGCC
z26f	GGCGTAAAATGcctACTGGTAATGC
z26r	GCATTACCAGTaggCATTTTACGCC
z27f	GGCGTAAAATGtttACTGGTAATGC
z27r	GCATTACCAGTaaaCATTTTACGCC
z28f	GGCGTAAAATGaatACTGGTAATGC
z28r	GCATTACCAGTattCATTTTACGCC
z29f	GGCGTAAAATGcagACTGGTAATGC
z29r	GCATTACCAGTctgCATTTTACGCC
z30f	GGCGTAAAATGaagACTGGTAATGC
z30r	GCATTACCAGTcttCATTTTACGCC
z31f	GGCGTAAAATGgatACTGGTAATGC
z31r	GCATTACCAGTatcCATTTTACGCC
z32f	GGCGTAAAATGgaaACTGGTAATGC
z32r	GCATTACCAGTttcCATTTTACGCC
L-23F	ACTGTCTGCTTACATAAACAGTAATACAAGGGGTGTTATG
L-23R	CATTAGTTTGCATTACCAGTggtCATTTTACGCCTCCTAT
L-24F	ATAGGAGGCGTAAAATGaccACTGGTAATGCAAACTAATG
L-24R	CATAACACCCCTTGTATTACTGTTTATGTAAGCAGACAGT
L-27F	CATTAGTTTGCATTACCAGTaaaCATTTTACGCCTCCTAT
L-27R	ATAGGAGGCGTAAAATGtttACTGGTAATGCAAACTAATG
L-34F	TTTGCATTACCAGTcggACGCATTTTACGCCTCCTATTA
L-34R	TAATAGGAGGCGTAAAATGCGTccgACTGGTAATGCAAA
L-35F	CATTAGTTTGCATTACCcggACGCATTTTACGCCTCCTAT
L-35R	ATAGGAGGCGTAAAATGCGTccgGGTAATGCAAACTAATG

^1^ Lowercase letters are mutated amino acids.

## Data Availability

The data are contained within the article.

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
