# Peer review of "Engineering and Purification of Microcin C7 Variants Resistant to Trypsin and Analysis of Their Biological Activity"

_antibiotics, 2023, doi:10.3390/antibiotics12091346_

Round 1

Reviewer 1 Report

It is interesting work could attract broad readership in the area of peptide and even protein engineering. Manuscript describe site directed mutagenisis of gentic varient which enhance antimicrobial and trypsine resistance. I love reading and reviewing your manuscript. I recommond manuscript for the publications in antibiotics. 

Author addresses site specific mutagenesis in microcin where desired amino acid can be replaced with new amino acids during downstream process of protein synthesis. This study shows they can improve trypsine resistances. Topic is relevant and suitable for antibiotic journals. site specific mutagenesis is important where desired amino acids can be replaced to imply  specific function in. the proteins.

I will be interested if the method is universal and can be applied to any gene to alter specific mutagenesis.

And conclusion is aligned with discussion and results in the manuscript. 

Table 6, should correct table 7 as there are 7 tables in the manuscript.

Reviewer 2 Report

The aim of this study was to engineer microcin C (McC) for enhanced trypsin resistance without loss of antibiotic activity, in order to improve performance against pathogens in the intestinal tract where McC is degraded by trypsin. This was achieved through mutation of the McC structural heptapeptide at and around residue R2, which is a known cleavage site for trypsin. The authors noted that this residue is also thought to be important for uptake of McC by target bacteria, and thus screened a wide selection of McC analogues to determine which retained McC antibacterial activity whilst displaying resistance to trypsin. The rationale for the study was clear and well-explained, and the authors were successful in achieving their aim. Some points/questions:

·       The reasons for mutation of the second arginine residue are obvious from the introduction, however, reasons behind the T3P and RPT mutations are not made clear until the discussion. It might be useful to the reader to touch on this earlier.

·       It would have been nice to see some quantitative evaluation of the trypsin sensitivity of the McC analogues. The amount of intact McC/McC analogues in the culture supernatants before and after trypsin treatment is unclear, and the halo sizes following treatment of all supernatants with trypsin are slightly reduced, indicating some degradation has occurred. The authors clearly have a method for purifying McC and its analogues. Treatment of the purified bacteriocins with trypsin and subsequent estimation of the amount of McC remaining, through mass spec analysis or adaption of the MIC assays, could strengthen their conclusions and enable better comparison of the trypsin resistance exhibited by different McC analogues if time allows.

·       What effect would the R2 mutations be expected have on re-uptake of McC and its analogues by the expression E. colistrain during fermentation, and would this be expected to affect the concentration of bacteriocin in the harvested supernatant thus the efficiency of trypsin digest?

·       Page 7, line 238 -241 – ‘Some previous experiments have studied the effects of peptide chain shortening on 238 YejABEF transporter recognition and antimicrobial activity and found that compared with wild-type McC, McC analogs containing 1 to 6 amino acids exhibited higher MIC values against E. coli K-12 (BW28357), which is similar to the trends of the MIC results in our study. I found this sentence a little confusing as the effect of chain length on the antimicrobial efficacy of the McC analogues is not addressed by this paper.

·       Page 8, line 253-255 – Another reason for this finding may be that McC also has a self-immune system. Because the induction capacity was un-changed, the variant was not recognized by the self-protection system’ – is depleted a better word to use here than recognised? As there is no evidence here to suggest that the mutations affect recognition by MccF, the self-immunity protein which recognises and degrades unprocessed McC.

·       Page 8, line 260-261 – ‘In this study, the T3P and RPT variants exhibited resistance to trypsin, and the results showed that these two variants lacked antimicrobial activity’ – What is the evidence for this?

The Authors produce their mutated MccA by secretion some of these mutants could be more or less secreted altering the inhibition phenotype.  this should be discussed.

If the introduction describe very well the potential of bacteriocins and the role they may play in industrial applications the authors in the discussion do not generalize their findings for other bacteriocins.  This work shown that bacteriocins Thanks to targeted mutations could be modulable and this represent a great potential.

Overal good quality of English.

Reviewer 3 Report

Microcins are one of the interesting candidate for generation of next generation antibiotics. However, there stability is a question. This paper tried to show a structure-activity relationship of McC7. My comments are below for this article.

1. English proofing is required.

2. Line 36, animal spelling?

3. Gene map and biosynthesis pathways of McC should be included in the manuscript.

4. Section 2.1, a result graph should be included.

5. What happen when you make a substitution at position 6? Both with and without the effective substitution at position 2?

6. What is RPT? Is P introduced between R and T? If yes, there will be change in the sequence length.

7. Digestion profile of peptide should be shown.

8. Supplementary figure 2 should have detailed description.

9. Description of K88 strain?

10. In line 154, what was the basis for 17 min assumption?

11. Stability of mutated peptides need to be checked with other digestive enzyme?

12. Is there any physical or chemical change in the characteristics of the mutated peptide?

13. What happen if we mutated the amino acids with D-amino acid? Or D-amino acids peptide would be more stable? Please comment.

14. Line 264, what is the reason behind R2P mutation?

English proofing is required.
